# Recombinant cyclin B-Cdk1-Suc1 capable of multi-site mitotic phosphorylation *in vitro*

**Keishi Shintomi** (ORCID)*, **Yuki Masahara-Negishi, Masami Shima**¤, **Shoji Tane** (ORCID), **Tatsuya Hirano** (ORCID)

Chromosome Dynamics Laboratory, RIKEN Cluster for Pioneering Research, Wako, Saitama, Japan

¤ Current address: Neural Circuit of Multisensory Integration RIKEN Hakubi Research Team, RIKEN Center for Brain Science, Wako, Saitama, Japan

* kshintomi@riken.jp

**Data Availability Statement:** All relevant data are within the paper.

**Funding:** K.S., Grant-in-Aid for Scientific Research, KAKENHI (grant numbers 19H05755 and

## Abstract

Cyclin-dependent kinase 1 (Cdk1) complexed with cyclin B phosphorylates multiple sites on hundreds of proteins during mitosis. However, it is not fully understood how multi-site mitotic phosphorylation by cyclin B-Cdk1 controls the structures and functions of individual substrates. Here we develop an easy-to-use protocol to express recombinant vertebrate cyclin B and Cdk1 in insect cells from a single baculovirus vector and to purify their complexes with excellent homogeneity. A series of *in-vitro* assays demonstrate that the recombinant cyclin B-Cdk1 can efficiently and specifically phosphorylate the SP and TP motifs in substrates. The addition of Suc1 (a Cks1 homolog in fission yeast) accelerates multi-site phosphorylation of an artificial substrate containing TP motifs. Importantly, we show that mitosis-specific multi-subunit and multi-site phosphorylation of the condensin I complex can be recapitulated *in vitro* using recombinant cyclin B-Cdk1-Suc1. The materials and protocols described here will pave the way for dissecting the biochemical basis of critical mitotic processes that accompany Cdk1-mediated large-scale phosphorylation.

## Introduction

For the accurate transmission of genomic information in eukaryotic cells, multiple mitotic events, such as nuclear envelope breakdown, chromosome condensation, and spindle assembly, must be completed in a tightly regulated manner. Such dynamic rearrangements of intracellular structures are under the control of cyclin-dependent kinase 1 (Cdk1), which is complexed with cyclin B [1, 2]. A rapid increase in Cdk1's activity from mitotic prophase to metaphase results in large-scale phosphorylation of a variety of protein substrates, often at multiple sites on individual substrates [3].

The activity of cyclin B-Cdk1 is regulated by multilayered mechanisms. By late G2 phase, cyclin B-Cdk1 is maintained in an inactive form in which the T14 and Y15 residues of Cdk1 are phosphorylated by the kinases Wee1 and Myt1, respectively. Upon entry into mitosis, these inhibitory modifications are removed by the protein phosphatase Cdc25, thereby converting cyclin B-Cdk1 into an active form [4, 5]. A CDK-activating kinase (CAK) is also responsible for full activation of cyclinB-Cdk1 by phosphorylating T161 in the T-loop of Cdk1,

22H02551), received from Japan Society for the Promotion of Science, https://www.jsps.go.jp/english/. T.H., Grant-in-Aid for Scientific Research, KAKENHI (grant numbers and 18H05276 and 20H05938), received from Japan Society for the Promotion of Science, https://www.jsps.go.jp/english/. The funder had no role in study design, data collection and analysis, decision to publish, or preparation of the manuscript.

**Competing interests:** The authors have declared that no competing interests exist.

which is located close to its catalytic center [6–8]. Finally, Cks1, a small adaptor protein that bridges Cdk1 and phosphorylated threonine-containing substrates, is known to accelerate multi-site phosphorylation reactions observed *in vivo* [3, 9–12].

Although the cell cycle regulation of Cdk1 has been extensively studied over the past decades, much less is known about how the structure and functions of Cdk1's substrates might be regulated by phosphorylation. To address this fundamental question, there is a high demand for technically simple purification of a pure and active fraction of cyclin B-Cdk1 and for recapitulation of physiologically relevant, multi-site phosphorylation reactions *in vitro*. Several protocols for the preparation of native and recombinant versions of cyclin B-Cdk1 have been published so far [8, 13–22]. However, they are not necessarily technically straight-forward, and the starting materials, if they are of native origin, may not be readily available to many researchers. In this paper, we report a modified protocol for the production of recombinant cyclin B-Cdk1 with high homogeneity. The resultant kinase can specifically phosphory-late cyclin-dependent kinase consensus motifs (SP and TP sites) of substrates [23] with a catalytic efficiency better than that of budding yeast recombinant M-CDK previously reported [24]. Furthermore, our *in vitro* assays demonstrate that mitosis-specific multi-site phosphory-lation of the condensin I complex, an essential player in chromosome condensation [25–27], can be recapitulated using recombinant cyclin B-Cdk1 and Suc1, a fission yeast homolog of Cks1 [28].

## Materials and methods

### Preparation of recombinant *X. tropicalis* and human M-CDKs

A cDNA fragment for *X. tropicalis* Cdk1 carrying T14A/Y15F mutations and flanked with a C-terminal hexahistidine (His) tag was codon-optimized for *Trichoplusia ni* and synthesized using commercial services provided by Eurofins Genomics. The fragment for *X. tropicalis* cyclin B1 (amino acids 131–397) carrying C133S/C142S/C316S mutations and flanked with an N-terminal 3×FLAG tag and a C-terminal Twin-Strep (TS) tag was synthesized in the same way. These fragments are individually cloned into between a polyhedrin promotor and termi-nator sequences on the plasmid vector pLIB (Addgene, Catalog # 80610) to create the gene expression cassettes (GECs). Next, a pair of the GECs are cloned into a pBIG1D vector (Addgene, 80614) according to the biGBac assembly methods [29, 30]. Using the resultant vec-tor, the *Escherichia coli* strain DH10EMBacY (Geneva Biotech) was transformed to generate a recombinant bacmid DNA via Tn7-mediated transposition. Baculoviruses were generated by introducing the bacmid DNAs into Sf9 cells (Thermo Fisher Scientific, 11496015) and ampli-fied by another round of infection. High-Five cells (Thermo Fisher Scientific, B85502) were then infected with the amplified viruses and grown in the Express Five media (Thermo Fisher Scientific, 10486025) at 27°C for 48 h. The cells were harvested and resuspended in buffer HisL (50 mM sodium phosphate [pH 7.4], 500 mM NaCl, 10 mM imidazole) supplemented with EDTA-free Complete Protease Inhibitor Cocktail (Merck, 11873580001) and the Benzo-nase nuclease (Merck, 71205), and lysed by sonication. The lysate was clarified by centrifuga-tion at 30,000 $g$ for 15 min and applied to a Ni$^{2+}$-charged Chelating Sepharose Fast Flow column (Cytiva, 17057501). The column was washed with buffer HisW (50 mM sodium phos-phate [pH 7.4], 500 mM NaCl, 50 mM imidazole), and bound proteins were eluted with buffer HisE (50 mM sodium phosphate [pH 7.4], 500 mM NaCl, 250 mM imidazole). For further purification, the eluate from the Ni$^{2+}$-column was dialyzed against buffer ST200 (20 mM Hepes-NaOH [pH 8.0], 200 mM NaCl, 10% glycerol), and loaded to a *Strep*-Tactin Sepharose column (IBA Lifesciences, 2-1201-010). The column was washed with buffer ST200, and bound proteins were eluted with buffer ST200 supplemented with 2.5 mM D-desthiobiotin

(Nacalai Tesque, 2-1000-005). Peak fractions were pooled, dialyzed against buffer KHG150/10 (20 mM Hepes-KOH [pH 7.7], 150 mM KCl, 10% glycerol) supplemented with 0.5 mM tris (2-carboxyethyl)phosphine (TCEP), and concentrated with an Amicon Ultra 30K-device (Merck, UFC503024). The concentrated protein sample was dispensed into small aliquots, snap-frozen in liquid nitrogen, and stored at -80˚C until use. Once an aliquot was thawed, it was stored on ice and used within 2 weeks. To evaluate the homogeneity of every preparation, an aliquot was analyzed by SDS-PAGE followed by Coomassie Brilliant Blue (CBB) staining. The concentration of the final preparation of M-CDK was determined by measuring the absorbance at 280 nm. For an analytical purpose, the M-CDK preparation purified from a *Strep*-Tactin-conjugated column was further fractionated by size-exclusion chromatography (SEC) using a Superose 6 increase 10/300 GL column (Cytiva, 29091596). To validate the presence of pT161-positive Cdk1 in the final preparation, immunoblotting was carried out using anti-pT161 and anti-PSTAIR antibodies, and images were acquired by using an Image analyzer (Odyssey XF, LI-COR Biosciences [for fluorescence detection] or Amersham Imager 680, Cytiva [for chemiluminescence detection]). Human recombinant M-CDK was produced using almost the same protocol. cDNA fragments for a full-length version of *Homo sapiens* Cdk1 carrying T14A/Y15F mutations (flanked with a C-terminal His tag) and an N-terminally deleted version of cyclin B1 (amino acids 165–433) carrying C167S/C238S/C350S mutations (flanked by an N-terminal 3×FLAG tag and a C-terminal Twin-Strep [TS] tag) were codon-optimized for *Trichoplusia ni* and synthesized (Eurofins Genomics). These cDNAs were cloned into multi-cloning sites downstream of the polyhedrin and p10 promoters on the pFastBac Dual vector (Thermo Fisher Scientific, 10712024). The resultant vector was used to transform the *Escherichia coli* strain DH10Bac (Thermo Fisher Scientific, 10361012) to generate a recombinant bacmid DNA. Baculovirus production, protein expression, and tandem affinity purification were performed as described above. A yield of the cyclin B-Cdk1 complex from 1 g insect cells was as follows: *X. tropicalis* M-CDK, 10~30 μg at the concentration of 1~2 μM; human M-CDK, 50~100 μg at the concentration of 2~10 μM.

### Preparation of recombinant proteins other than M-CDK

**Linker histones.** cDNAs encoding of *X. laevis* H1.1 and H1.8 (provided by Kiyoe Ura [Chiba University, Japan] and Keita Ohsumi [Nagoya University, Japan], respectively) were cloned into the pET28-3C vector (a lab-made derivative of pET28a [Novagen, 69864], in which the original thrombin protease recognition site was replaced with the 3C protease recognition site). A cDNA fragment for *X. laevis* H1.1 carrying S171A/S184A/S197A/S213A/S232A mutations (namely, all five serine residues of the SP motifs have been substituted with alanines; hereafter referred to as the 5A mutant) was synthesized (Eurofins Genomics) and cloned into the pET28-3C vector. Both H1.1 (wild-type and 5A) and H1.8 were expressed and purified by the same procedure. The plasmid DNA was introduced into the *E. coli* strain BL21 (DE3), and the transformant was grown in LB at 37˚C until $OD_{600}$ reached 0.8. IPTG was then added at a final concentration of 200 μM. After an 1-hr incubation at 37˚C, the culture was transferred to a temperature of 20˚C, and incubation was continued for another 16 hr to allow the expression of the recombinant proteins. The cells were harvested, suspended in buffer HisL supplemented with EDTA-free Complete Protease Inhibitor Cocktail (Roche), and lysed by sonication. The lysate was clarified by centrifugation and loaded on a $Ni^{2+}$-charged Chelating Sepharose Fast Flow column. After washing the column with buffer HisW, the bead-binding proteins were eluted with buffer HisE. The eluate was dialyzed against buffer S100 (20 mM HEPES-NaOH [pH 7.2], 100 mM NaCl) along with lab-made human rhinovirus 3C protease to liberate the His tag and applied to a HiTrap SP HP column (Cytiva, 17115101). The column was developed

with a linear gradient of NaCl (100–600 mM). Peak fractions were pooled and dialyzed against buffer NHG150/10 (20 mM Hepes-NaOH [pH 7.7], 150 mM NaCl, 10% glycerol). Typically, approximately 1 mg of linker histones were obtained from a 1-liter culture and the concentration of stock aliquots ranged from 200 μM to 500 μM.

**XD2-C.** cDNA fragments encoding XD2-C WT and 3A were codon-optimized for *E. coli* and synthesized (Eurofins Genomics) and cloned into the pMAL-c6T vector (New England Biolabs, N0378). Protein expression in the *E. coli* host was performed as described above. The resultant XD2-C proteins fused with a His-MBP tandem tag at their N-termini were purified using a $Ni^{2+}$-charged Chelating Sepharose Fast Flow column. Peak fractions were pooled and dialyzed against buffer KHG150/10. Typically, 300 μg of XD2-C were obtained from a 1-liter culture and the concentration of stock aliquots ranged from 50 μM to 100 μM.

**Suc1.** A cDNA encoding *Schizosaccharomyces pombe* Suc1 (provided by Eiichi Okumura [Tokyo Institute of Technology, Japan]) was cloned into the pET28-3C vector. Protein expression in the *E. coli* host and coarse purification using a $Ni^{2+}$-charged column were performed as described above. The resultant suc1 fused with a His tag at their N-terminus was purified by using a $Ni^{2+}$-charged Chelating Sepharose Fast Flow column. Peak fractions were pooled, further purified with a HiTrap Q HP column (Cytiva, 17115301), and dialyzed against buffer KHG150/10. Typically, 10 mg protein was obtained from a 1-liter culture and the concentration of stock aliquots ranged from 200 μM to 500 μM.

**Condensin I holocomplexes.** A baculovirus vector for the expression of *X. laevis* condensin I holocomplex was made according to the biGBac assembly protocol [29, 30]. Briefly, cDNA fragments encoding five subunits of condensin I (XCAP-C/Smc4, -D2, -E/Smc2, -G, -H [fused with a Twin-Strep tag at its C-terminus]) were codon-optimized for *Trichoplusia ni* and synthesized (GeneArt, Thermo Fisher Scientific). They were inserted stepwise into intermediate plasmid vectors (namely, pLIB and pBIG1 series vectors) and finally combined into a single pBIG2ABC vector (Addgene, 80617). Baculovirus production and protein expression were carried out as described for M-CDKs. The resultant condensin I holocomplex was first purified with a *Strep*-tactin-conjugated column and further purified by SEC using a Superose 6 increase column. Peak fractions were pooled and dialyzed against buffer KHG150/10. Typically, 20 μg of the protein complex was obtained from 1 g insect cell pellet at a concentration of 2 μM. A mammalian version of the condensin I holocomplex (composed of mouse Smc2 and Smc4, and human CAP-D2, -G, and -H) was prepared as previously described [31, 32].

## Phosphorylation assays using recombinant M-CDKs

A protein mixture containing a substrate (linker histone, XD2-C, or condensin I), M-CDK, and Suc1 (in a volume ranging from 30 to 100 μl) was dialyzed against kinase buffer (20 mM HEPES-KOH [pH 7.7], 80 mM KCl and 5 mM $MgCl_2$) using an Xpress Micro Dialyzer MD100 (Scienova, 40078) at 4°C. The dialysate was supplemented with ATP at a concentration of 2 mM and incubated at 25°C. The concentrations of protein components in each assay, which were determined by preliminary titration experiments and technical considerations, are listed in Table 1. At the indicated time points, aliquots were taken and mixed with an equal volume of 2×SDS sample buffer (125 mM Tris-HCl [pH 6.8], 10% 2-mercaptoethanol, 4% SDS, 20% glycerol, 0.2% Bromophenol blue). The resultant samples were subjected to SDS-PAGE. For analyzing either H1.1 or XD2-C by Phos-tag SDS-PAGE, gels containing 10% or 7.5% acrylamide, 50 μM Phos-tag acrylamide (Fujifilm Wako Pure Chemical, AAL-107), and 100 μM $MnCl_2$ were used, respectively. For visualizing phosphorylated proteins prior to CBB staining, the gels were stained with Pro-Q Diamond phosphoprotein stain gel solution (Thermo Fisher Scientific, P33300) according to the manufacturer's instruction with the

**Table 1. Concentrations of the protein components in various phosphorylation assays.**

| | Fig 2 | | | | Fig 3 | Fig 4 |
|---|---|---|---|---|---|---|
| | **B, C** | **D, E** | **F** | **G** | **B-E** | **A-C** |
| M-CDK [nM] | 50 | 10 | 10 | 10, 30, 60 | 20 | 50 |
| Substrate [μM] | 10 | 10 | 0.63~10 | 10 | 2 | 0.5 |
| Suc1 [μM] | - | - | | - | 0, 0.1, 1 | 0, 1 |
| | S1 Fig | | | | S2 Fig | |
| M-CDK [nM] | 10 | | | | 20 | |
| Substrate [μM] | 10 | | | | 2 | |
| Suc1 [μM] | - | | | | 0, 1 | |

exception that 1,2-propanediol was used instead of acetonitrile in the destaining step. After staining the gel, images were acquired using an image analyzer (Amersham Imager 680, Cytiva). Semi-quantitative analyses of the phosphorylation on H1.1 or XD2-C by M-CDK were performed as follows: First, average numbers of phosphorylated residues per each substrate molecule were calculated at each time point by quantifying band density in Phos-tag gels and plotted against the time (Figs 2E and 3E, kinetics [left panels]); Next, the slope of the abovementioned kinetics graph for the first 10 minutes was calculated, and the value multiplied by the molar ratio of the substrate to M-CDK was approximated as the phosphorylation rate (Figs 2E and 3E, rate [right panels]). An enzymatic kinetics analysis of M-CDK was performed as follows. First, increasing concentrations of H1.1 (equivalent to the concentration of the SP sites ranging from 3.125 to 50 μM) were phosphorylated by a fixed concentration of M-CDK (10 nM). Second, the concentrations of phosphorylated SP sites generated during an initial 10-min incubation were quantified as described above, and then the resultant values were divided by the elapsed time (10 min) to approximate the initial velocity of the reaction ($v$). Third, the $v$ values obtained from three independent experiments were plotted against the concentration of the SP sites ([S]) (Fig 2F). Finally, the values of $V_{max}$ and $K_m$ (the Michaelis constant) were estimated by fitting the data to the Michaelis-Menten equation $v = V_{max} / (1 + (K_m / [S]))$ using the PRISM software (GraphPad), and the $k_{cat}$ (the rotation number) was given by the quotient of the $V_{max}$ value divided by the initial concentration of M-CDK (0.010 μM).

## Antibodies

Primary antibodies used in the current study were as follows: anti-XCAP-C (in-house identifier: AfR8) [33]; anti-CAP-E (in-house identifier: AfR9) [33]; anti-XCAP-D2 (in-house identifier: AfR16) [25]; anti-XCAP-H (in-house identifier: AfR20) [25]; anti-pT1314-XCAP-D2 (in-house identifier: AfR42-P) [25]; anti-pT1348-XCAP-D2 (in-house identifier: AfR44-P) [25]; anti-pT1353-XCAP-D2 (in-house identifier: AfR46-P) [25]; anti-hCAP-D2 (in-house identifier: AfR51-5L) [34]; anti-pT1339-hCAP-D2 (in-house identifier: AfR173-4P) [32]; anti-pT1384-hCAP-D2 (in-house identifier: AfR175-4P) [32]; anti-FLAG M2 (Sigma, F-1804 [RRID: AB_262044]); anti-PSTAIR (Abcam, ab10345, [RRID: AB_297080]); anti-pT161-Cdk1 (Cell Signaling Technology, 9114, [RRID: AB_2074652]). Secondary antibodies used in the current study were as follows: horseradish peroxidase-conjugated anti-rabbit IgG (Vector Laboratories, PI-1000 [RRID: AB_2336198]); horseradish peroxidase-conjugated anti-mouse IgG (Vector Laboratories, PI-2000 [RRID: AB_2336198]); IRDye 680RD goat anti-rabbit IgG (LI-COR Biosciences, 925–68071 [RRID: AB_2721181]); IRDye 800CW goat anti-mouse IgG (LI-COR Biosciences, 925–32210[RRID: AB_2687825]).

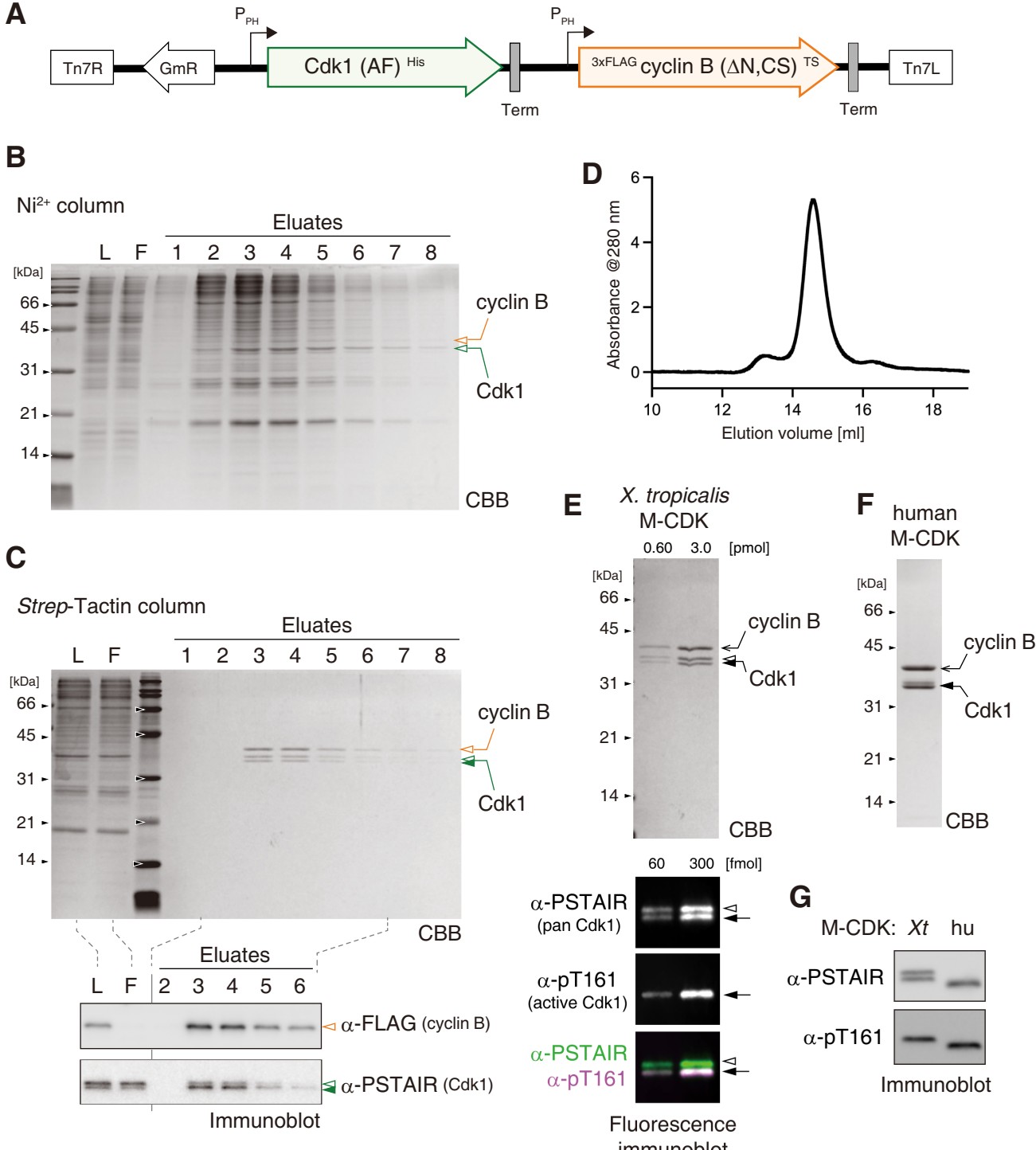

**Fig 1. Preparation of recombinant cyclin B-Cdk1 complexes.** (**A**) cDNA fragments encoding engineered versions of *Xenopus tropicalis* Cdk1 and cyclin B were cloned into a plasmid vector. The DNA segment depicted in the cartoon was transposed into a single baculovirus vector, and the two polypeptides were simultaneously expressed in insect cells. (**B**) A lysate was prepared from the virus-infected cells and fractionated using a $Ni^{2+}$-charged column. The loaded sample (L), the flow-through (F), and the eluates were analyzed by SDS-PAGE and stained with Coomassie Brilliant Blue (CBB). (**C**) Eluates from the $Ni^{2+}$-charged column were further fractionated using a *Strep*-Tactin-conjugated column. Samples were analyzed by SDS-PAGE followed by CBB staining (top). Selected samples were also analyzed by immunoblotting with the indicated antibodies (bottom). (**D**) The cyclin B-Cdk1 complex purified by tandem affinity chromatography was analyzed by size-exclusion chromatography (SEC). Its elution profile monitored by absorbance at 280 nm is shown. (**E**) The final preparation of *X. tropicalis* M-CDK was analyzed by SDS-PAGE followed by CBB staining (top). The same set of samples was

analyzed by fluorescence immunoblotting (bottom) with anti-pT161 (to visualize an active form of Cdk1) and anti-PSTAIR (to visualize both active and inactive form of Cdk1). (**F**) The final preparation of human M-CDK was analyzed by SDS-PAGE. (**G**) *X. tropicalis* and human M-CDKs were analyzed side-by-side by immunoblotting with the indicated antibodies.

## Results and discussion

### Preparation of recombinant cyclin B-Cdk1 complexes expressed in insect cells

We wished to produce a homogeneous and enzymatically active preparation composed of recombinant cyclin B and Cdk1 for the functional and structural characterization of Cdk1's substrates. To this end, we designed a baculoviral vector by taking into account the following points: (1) to stabilize the cyclin B moiety in host insect cells, the N-terminal region of *Xenopus tropicalis* cyclin B1 (amino acids 1–130), which is required for ubiquitin-mediated proteolysis [35], was deleted, and three non-conserved cysteines (i.e., C133, C142, and C316) were replaced with serines as previously shown [36]; (2) to prevent Cdk1 from receiving inhibitory phosphorylations in the host cells, T14 and Y15 in *X. tropicalis* Cdk1 were mutated to alanine and phenylalanine, respectively; (3) to maximize the yield and homogeneity of the purified protein complex, these mutant proteins (cyclin B1 (ΔN, CS) and Cdk1 (AF)) were conjugated with affinity tags (3×FLAG-tag and Twin-Strep [TS]-tag at the N- and C-termini of cyclin B1 and the hexahistidine [His]-tag at the C-terminus of Cdk1), and were expressed from a single vector rather than from a combination of vectors expressing individual proteins [29, 30] (Fig 1A).

From High-Five insect cells infected with the recombinant virus, the cyclin B-Cdk1 complex was successfully purified by tandem affinity chromatography using $Ni^{2+}$ and *Strep*-Tactin columns (Fig 1B and 1C). Size-exclusion chromatography of the eluate from the second column resulted in a single major peak, demonstrating that our two-step purification protocol yielded a protein preparation mainly composed of one molecule each of cyclin B1 (ΔN, CS) and Cdk1 (AF) (Fig 1D). We noticed that Cdk1 in the purified preparation was separated into two discrete bands as judged by SDS-PAGE. Fluorescence immunoblotting analysis, which allowed us to simultaneously detect pan-Cdk1 (recognized by anti-PSTAIR) and its active form (recognized by anti-pT161) in different channels, demonstrated that the faster migrating band corresponds to the active form of Cdk1 (Fig 1E). It is reasonable to speculate that endogenous CAK phosphorylates T161 of the recombinant Cdk1 in the host insect cells. We also engineered human cyclin B1 and Cdk1 sequences in a similar strategy and succeeded in producing a homogeneous and pT161-positive recombinant complex (Fig 1F and 1G, see Materials and Methods for details). We hereafter refer to the purified cyclin B-Cdk1 complex as M-CDK, which stands for <u>m</u>itotic <u>c</u>yclin-<u>d</u>ependent <u>k</u>inase, and show the results using *X. tropicalis* M-CDK unless otherwise indicated.

### Recombinant M-CDKs phosphorylate a substrate containing SP motifs

To clarify whether the recombinant M-CDKs are enzymatically active and, if so, to what extent, we decided to use two different variants of recombinant *Xenopus laevis* linker histones as substrates: one was H1.1, which has five SP motifs, and the other was H1.8, which has no SP or TP motifs (Fig 2A). We first set up a reaction mixture, in which recombinant H1.1 or H1.8 (10 μM) was mixed with M-CDK (50 nM) along with ATP (2 mM) (Table 1). Aliquots of the reaction mixture were taken at various time points and subjected to conventional SDS-PAGE. In the resulting gel, phosphorylated proteins were visualized with the Pro-Q Diamond solution, and then total proteins were stained with CBB. The result clearly demonstrated that H1.1

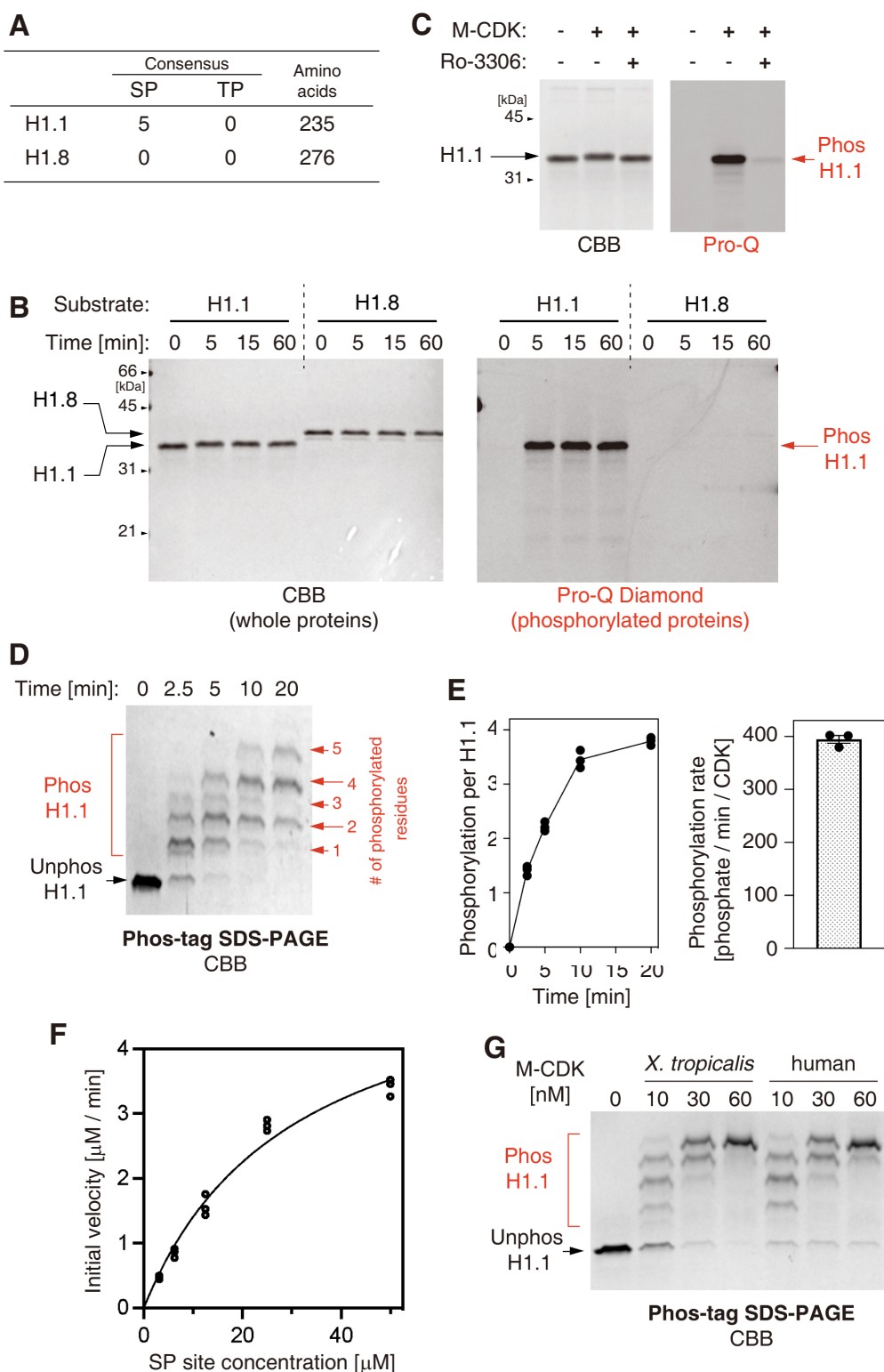

**Fig 2. Recombinant M-CDKs phosphorylate a substrate containing SP motifs.** (**A**) The *Xenopus laevis* linker histone variant H1.1 contains five SP and no TP motifs whereas another variant H1.8 contains neither SP nor TP motifs. (**B**) Recombinant H1.1 and H1.8 were incubated with *X. tropicalis* M-CDK and ATP at 25°C. At the indicated time points, the reactions were terminated and analyzed by SDS-PAGE. The gel was first stained with Pro-Q Diamond solution to visualize phosphorylated proteins and then stained with CBB to visualize total proteins. (**C**) An H1.1

phosphorylation assay was performed in the absence or presence of the Cdk1-specific inhibitor Ro-3306. After a 60-min incubation, the reactions were analyzed as above. (**D**) H1.1 was phosphorylated with a relatively low concentration of M-CDK (see Table 1). At the indicated time points, the reactions were analyzed by Phos-tag SDS-PAGE to separate proteins phosphorylated at different levels into discrete bands. (**E**) Kinetics (left) and rate (right) of H1.1 phosphorylation were quantified. The mean ± standard errors from three independent experiments are shown. (**F**) Increasing concentrations of H1.1 were phosphorylated with a fixed concentration of M-CDK. The initial velocities of phosphorylation in each reaction were plotted against the concentrations of SP sites (equivalent to five times the concentration of H1.1) and fitted to the Michalis-Menten equation (which is indicated by a regression curve). The estimated kinetic parameters are shown in Table 2. (**G**) H1.1 was phosphorylated with increasing concentrations of *X. tropicalis* and human M-CDKs. After a 10-min incubation, the reactions were terminated and analyzed by Phos-tag SDS-PAGE.

was rapidly phosphorylated by purified M-CDK whereas H1.8 was hardly phosphorylated under the same condition (Fig 2B). We also confirmed that the phosphorylation of H1.1 was blocked when the Cdk1-specific inhibitor Ro-3306 was added to the reaction mixture [37] (Fig 2C).

We next investigated the kinetics of H1.1 phosphorylation in more detail by using a Phos-tag acrylamide-containing gel that magnifies the mobility shift of phosphorylated proteins [38]. When a relatively low concentration of M-CDK (10 nM) was added to the assay (Table 1), phosphorylation of H1.1 gradually proceeded up to 20 min (Fig 2D). Populations of H1.1 phosphorylated at 1–2 sites were dominant at early time points, whereas multiple (up to five) sites of H1.1 were phosphorylated sites at late points. In addition, an H1.1 mutant (5A), in which all five serine residues of the SP motifs (S171, S184, S197, S213, and S232) had been replaced with alanines, was hardly phosphorylated in the same setup (S1 Fig). Taken together, it is reasonable to claim that our recombinant M-CDK catalyzes phosphorylation selectively at the SP sites of H1.1 that match the cyclin-dependent kinase consensus sites [23]. However, we cannot exclude the possibility that any one of the five sites was not phosphorylated in our assay. Semi-quantitative estimation showed that each M-CDK molecule catalyzes the phosphorylation of H1.1 at a rate of 394 ± 7.5 [phosphate/min] (mean ± standard error)(Fig 2E). To further validate the enzymatic property, we estimated the kinetic parameters of the M-CDK for the H1.1 substrate using a Michaelis-Menten plot. The result demonstrates that *X. tropicalis* M-CDK developed here catalyzes phosphorylation reactions with an efficiency (the values of $k_{cat} / K_m$) superior or comparable to that of budding yeast or human recombinant M-CDK (Fig 2F and Table 2) [8, 24]. We also confirmed that our preparation of human M-CDK could phosphorylate H1.1 and that its activity was roughly equivalent to that of *X. tropicalis* M-CDK (Fig 2G).

## Suc1 accelerates M-CDK phosphorylation of a substrate containing multiple TP motifs

Condensin I is a pentameric protein complex that plays an indispensable role in mitotic chromosome assembly [39, 40]. A native condensin I complex purified from egg extracts has been shown to act as one of the six protein factors required for chromosome reconstitution *in vitro*,

**Table 2. Kinetic parameters of *X. tropicalis* M-CDK and their comparison with previous studies.**

| Kinase | Substrate | $K_m$ | $k_{cat}$ | $k_{cat} / K_m$ | Reference |
|---|---|---|---|---|---|
| | | [μM] | [min$^{-1}$] | [min$^{-1}$ μM$^{-1}$] | |
| *X. tropicalis* cyclin B1-Cdk1 | H1.1 | 30.2 | 565 | 18.7 | This study |
| *S. cerevisiae* Clb2-Cdc28 | H1 (peptide) | 76.4 | 313 | 4.1 | [24] |
| *H. sapiens* cyclin B1-Cdk1 | Synthetic peptide | Not shown | Not shown | 31.2 | [8] |

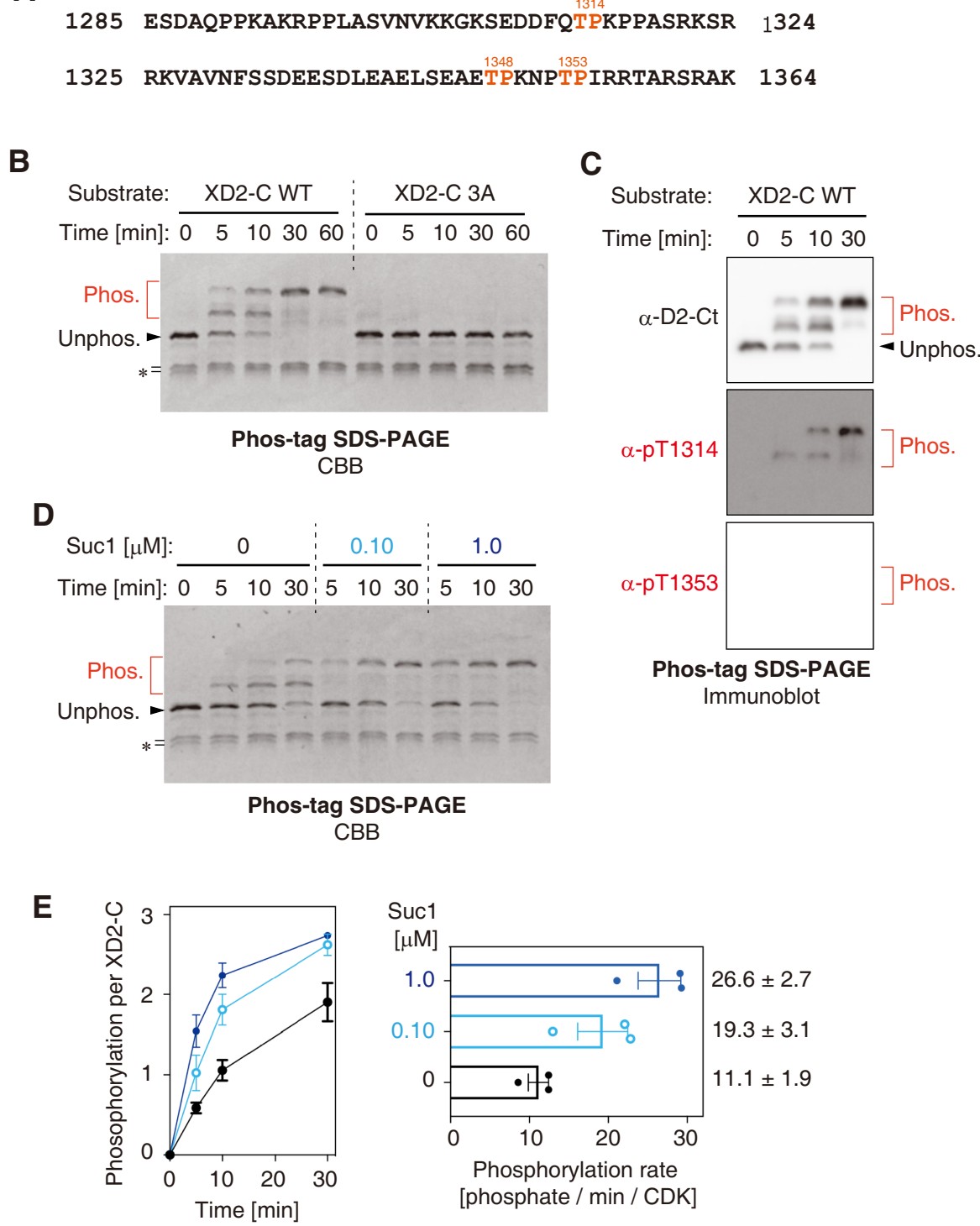

**Fig 3. Suc1 accelerates M-CDK phosphorylation of a substrate containing multiple TP motifs.** (**A**) Primary structure of the C-terminal region of XCAP-D2 (amino acids 1285–1364, XD2-C) that was expressed in *E. coli* as a fusion with maltose-binding protein (MBP). XD2-C contains three TP motifs (shown in red). In XD2-C 3A mutant, all threonine residues in these motifs were replaced with alanines. (**B**) XD2-Cs (wild-type [WT] and 3A) were incubated with M-CDK and ATP. Aliquots were taken at the indicated time points and subjected to Phos-tag SDS-PAGE followed by CBB staining. The asterisk (*) indicates a degradation product of XD2-C. (**C**) Selected reactions from an XD2-C phosphorylation assay (shown in panel B) were subjected to Phos-tag SDS-PAGE followed by immunoblotting. (**D**) XD2-C was phosphorylated by M-CDK in the presence of different concentrations of Suc1 and analyzed as described in (B). (**E**)

Kinetics (left) and rate (right) of XD2-C phosphorylation were quantified. The mean ± standard errors from three independent experiments are shown.

but it does so only when it is phosphorylated by cyclin B-Cdk1 [27]. A number of Cdk1 phosphorylation sites in the condensin I subunits have been identified so far. Among them are three threonine residues (T1314, T1348, and T1353) in the C-terminal intrinsically disordered region (IDR) of the XCAP-D2 subunit in *X. laevis* [26] (Fig 3A). We tested whether these residues could be phosphorylated using our recombinant M-CDK. To this end, the C-terminal IDR of XCAP-D2 (referred to as XD2-C WT) was expressed as a polypeptide fused to the maltose-binding protein, purified and incubated with M-CDK as a substrate. We found that M-CDK efficiently phosphorylated this fusion protein, causing mobility shifts on a Phos-tag gel (Fig 3B) and generating phosphoepitopes (pT1314 and pT1353) as judged by immunoblotting (Fig 3C). In striking contrast, virtually no phosphorylation was detectable when a mutant fusion protein (XD2-C 3A), in which all three threonines (T1314, T1348, and T1353) were substituted with alanines, was used as a substrate (Fig 3B). These results convincingly demonstrated that the three TP motifs in XD2-C WT are the major targets of M-CDK.

The Cks1 family proteins are known to guide a substrate containing previously phosphorylated threonines to the active site of Cdk1, thereby promoting phosphorylation of other proximal sites [9]. We next tested whether Suc1, a fission yeast homolog of Cks1 [28], could facilitate the multi-site phosphorylation of XD2-C and, if so, to what extent the reaction was accelerated. We found that Suc1 indeed accelerated the phosphorylation of XD2-C (Fig 3D). Quantitation of Phos-tag SDS-PAGE revealed that Suc1 increased the rate of XD2-C phosphorylation approximately 2.4-fold under the assay conditions tested (Fig 3E). Furthermore, the 3A mutant of XD2-C was barely phosphorylated by our recombinant M-CDK even in the presence of Suc1, indicating that it does not phosphorylate additional sites other than the three TP sites in XD2-C, at least under the conditions tested (S2 Fig).

## Multi-site phosphorylation of the condensin I holocomplex can be recapitulated by M-CDK and Suc1

We next sought to recapitulate multi-subunit and multi-site phosphorylation of the condensin I holocomplex by M-CDK *in vitro*. To this end, a recombinant holocomplex composed of *X. laevis* condensin I subunits was expressed in insect cells, purified, and incubated with M-CDK and ATP. The resulting phosphorylation of the condensin I subunits was analyzed by SDS-PAGE, followed by Pro-Q diamond staining. We found that M-CDK alone phosphorylated the XCAP-C, -D2, and -H subunits to a modest level. Remarkably, inclusion of Suc1 in the reaction mixture resulted in much more efficient phosphorylation of the same set of subunits (Fig 4A). Notably, only in the presence of Suc1, XCAP-H displayed a massive mobility shift (Fig 4A and 4B), a phenomenon that had also been observed in mitotic egg extracts [25]. Moreover, Suc1 boosted the phosphorylation at T1314 and T1353 of XCAP-D2 in the holocomplex (Fig 4B), as well as that of the XD2-C substrate (Fig 3D and 3E).

Our recent work has shown that several SP and TP motifs in a recombinant mammalian condensin I complex are phosphorylated in mitotic egg extracts [32]. We confirmed that our M-CDK can phosphorylate at least two of them (T1339 and T1353 of human CAP-D2) *in vitro* (Fig 4C), thereby extending our above-mentioned observation using *X. laevis* condensin I as a substrate. Taken together, it is reasonable to conclude that the multi-site phosphorylation of condensin I subunits observed in mitotic egg extracts can be recapitulated with the recombinant proteins *in vitro*.

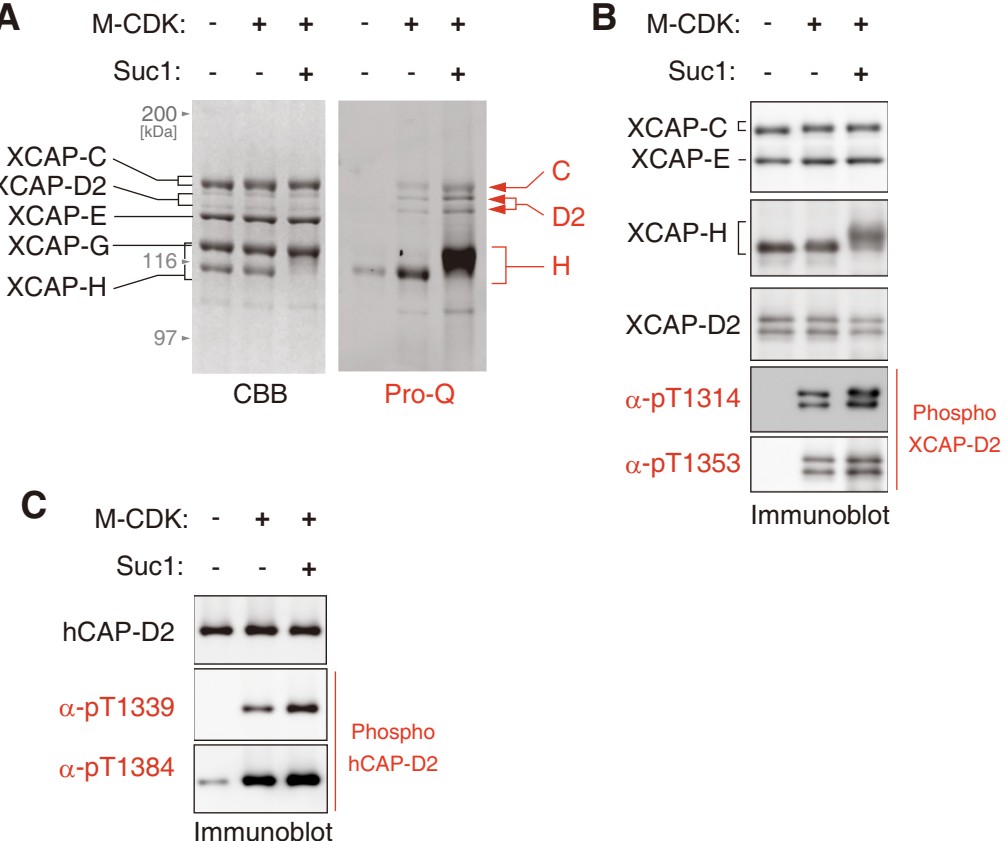

**Fig 4. Multi-site phosphorylation of the condensin I holocomplex can be recapitulated by M-CDK and Suc1.** (**A** and **B**) A recombinant *Xenopus laevis* condensin I holocomplex was phosphorylated by M-CDK with or without Suc1. The reaction mixtures were analyzed by SDS-PAGE. To visualize phosphorylated proteins, the gel was stained with Pro-Q Diamond solution (A). Mobility shifts of the XCAP-C and -H subunits and the generation of mitosis-specific phospho-epitopes (pT1314 and pT1353) on the XCAP-D2 subunit were examined by immunoblotting (B). (**C**) A recombinant mammalian condensin I holocomplex was phosphorylated by M-CDK with or without Suc1. The reaction mixtures were analyzed by immunoblotting with antibodies raised against mitosis-specific phospho-epitopes (pT1339 and pT1384) on the hCAP-D2 subunit.

## Conclusions and perspectives

In the current study, we describe a protocol for the production of frog and human versions of the recombinant cyclin B-Cdk1 complex (M-CDK) (Fig 1). It should be emphasized that our protocol is superior to previously reported ones in the following three points: (1) the recombinant M-CDKs can be easily and reproducibly purified with a hitherto unprecedented degree of homogeneity by two chromatography steps; (2) the phosphorylation at T161 of Cdk1, which is required for kinase activation, can be achieved in the host cells without the co-expression of exogenous CAK; (3) all procedures are easy to implement for standard molecular biology laboratories, and the expression vectors described in the current study are available upon request.

Furthermore, our preparation of *X. tropicalis* M-CDK exhibits a catalytic efficiency superior or comparable to that of previously reported recombinant M-CDK complexes (Table 2). We provide a compelling set of data that the recombinant M-CDKs specifically phosphorylate SP and TP motifs in H1.1 and XD2-C substrates (Figs 2 and 3, S1 and S2 Figs). However, our data do not exclude the possibility that Cdk1 also phosphorylates non-SP/TP sites in various cellular contexts [41, 42]. In the final set of assays, multi-subunit and multi-site phosphorylation of

the pentameric condensin I complex can be recapitulated by combining recombinant M-CDK with its cofactor Suc1. To our knowledge, this is the first report showing that physiologically relevant phosphorylation of condensin I can be achieved using recombinant M-CDK (Fig 4). Thus, recombinant M-CDKs developed in the current study will be of great help in the *in-vitro* reconstitution of elaborate protein machineries that govern critical processes in mitosis.

## Supporting information

**S1 Fig. Recombinant M-CDK specifically phosphorylates SP motifs in H1.1.** Wild-type (WT) and the 5SA mutant of H1.1 were incubated with M-CDK and ATP at 25˚C. Aliquots were taken at the indicated time points and subjected to conventional SDS-PAGE. The gel was first stained with Pro-Q Diamond solution to visualize phosphorylated proteins and then stained with CBB to visualize total proteins (a pair of upper panels). The same set of samples was also subjected to Phos-tag SDS-PAGE (a lower panel).
(EPS)

**S2 Fig. Recombinant M-CDK barely phosphorylate sites other than the three TP motifs in XD2-C with or without Suc1.** XD2-C WT and 3A were incubated with M-CDK and ATP in the absence or presence of Suc1 (1.0 μM) at 25˚C for 60 min. The resultant mixtures were subjected to SDS-PAGE (Pro-Q and CBB staining) and Phos-tag SDS-PAGE (CBB staining). The same set of samples was analyzed by immunoblotting using antibodies raised against the three phosphorylated TPs (pT1314, pT1348, and pT1353) and a C-terminal peptide (D2-Ct: residues 1351–1364). The asterisk (*) indicates a degradation product of XD2-C.
(EPS)

**S1 Raw images. Original uncropped images for blots and gels.**
(PDF)

## Acknowledgments

We thank K. Ura, K. Ohsumi, and E. Okumura for the reagents and members of the Hirano laboratory for their discussion and comments.

## Author Contributions

**Conceptualization:** Keishi Shintomi.

**Data curation:** Keishi Shintomi.

**Formal analysis:** Keishi Shintomi.

**Funding acquisition:** Keishi Shintomi, Tatsuya Hirano.

**Investigation:** Keishi Shintomi, Yuki Masahara-Negishi, Masami Shima, Shoji Tane.

**Methodology:** Keishi Shintomi, Yuki Masahara-Negishi, Masami Shima, Shoji Tane.

**Project administration:** Keishi Shintomi, Tatsuya Hirano.

**Resources:** Keishi Shintomi, Yuki Masahara-Negishi, Masami Shima, Shoji Tane.

**Supervision:** Keishi Shintomi, Tatsuya Hirano.

**Validation:** Keishi Shintomi.

**Visualization:** Keishi Shintomi.

**Writing – original draft:** Keishi Shintomi, Tatsuya Hirano.

**Writing – review & editing:** Keishi Shintomi, Tatsuya Hirano.

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
