## [Decision Letter · Decision Letter 0]

23 Oct 2023

PONE-D-23-30585Recombinant cyclin B-Cdk1-Suc1 capable of multi-site mitotic phosphorylation in vitroPLOS ONE

Dear Dr. Shintomi,

Thank you for submitting your manuscript to PLOS ONE. After careful consideration, we feel that it has merit but does not fully meet PLOS ONE’s publication criteria as it currently stands. Therefore, we invite you to submit a revised version of the manuscript that addresses the points raised during the review process.

I am willing to publish your manuscript, but it needs to be improved. Please answer all reviewers comments and specifically

1 - Better explain why your in vitro assay is better than existing assays; also find a way to demonstrate that the data obtained recapitulates certain in vivo situations by choosing the right substrates (see reviewer's comments). Establishing the classical kinetic parameters of a kinase would also be useful for comparison with other methods (see reviewer comment)

2 - Please explain why not use proteins from a single organism instead of using proteins from X. Tropicalis and human proteins for M-CDK with yeast Suc1.

3 - Please explain the use of different concentrations of M-CDK.

4 - Regarding multisite phosphorylation, please find a way to unambiguously control the identity of all phosphorylated sites (mutants or mass spectrometry) in H1.1 and XCAP-D2.

We look forward to receiving your revised manuscript.

Kind regards,

Claude Prigent

Academic Editor

PLOS ONE

[ We thank K. Ura, K. Ohsumi, and E. Okumura for the reagents and members of the Hirano 

laboratory for their discussion and comments. This work was supported by Grant-in-Aid for 

Scientific Research, KAKENHI (grants 19H05755 and 22H02551 [to K.S.], and 18H05276 and

20H05938 [to T.H.])]

 [K.S., Grant-in-Aid for Scientific Research, KAKENHI (grant numbers 19H05755 and 22H02551), received from Japan Society for the Promotion of Science, https://www.jsps.go.jp/english/.

T.H., Grant-in-Aid for Scientific Research, KAKENHI (grant numbers  and 18H05276 and 20H05938), received from Japan Society for the Promotion of Science, https://www.jsps.go.jp/english/.

The funder had no role in study design, data collection and analysis, decision to publish, or preparation of the manuscript.]

Reviewers' comments:

Reviewer's Responses to Questions

**Comments to the Author**

1. Is the manuscript technically sound, and do the data support the conclusions?

Reviewer #1: Yes

Reviewer #2: Partly

2. Has the statistical analysis been performed appropriately and rigorously? 

Reviewer #1: Yes

Reviewer #2: Yes

3. Have the authors made all data underlying the findings in their manuscript fully available?

Reviewer #1: Yes

Reviewer #2: Yes

4. Is the manuscript presented in an intelligible fashion and written in standard English?

Reviewer #1: Yes

Reviewer #2: Yes

5. Review Comments to the Author

Reviewer #1: The authors present an essentially methodological paper for the simple expression and purification of recombinant Xenopus and human cdk1-cyclin B complexes. They demonstrate that the kinases are active and can phosphorylate a recombinant condensin complex, and that addition of a small subunit (a CKS1 homologue) increases the phosphorylation rate. There are no particular surprises in these findings but the methodology and results are of high quality and the tools generated may be useful for the community. It would be good if the authors provided a bit of basic extra characterisation to support their claim that the method is superior to previous methods - namely, by establishing kinetic parameters of the purified kinases (Km and Vmax for ATP and a substrate, e.g. histone H1), which would also allow to see if there are differences between the Xenopus and human complexes. I also feel that to support the claim of efficient multisite phosphorylation of condensin complex, mass spectrometry should have been performed to identify phosphorylation sites. This is likely to also reveal that the recombinant kinases phosphorylate additional sites other than SP and TP sites, and might show whether Suc1 increases the rate of phosphorylation or allows additional sites to be phosphorylated.

Reviewer #2: In this study, Shintomi and colleagues develop a protocol to purify recombinant Cdk1-Cyclin B complexes from different organisms (Xenopus and human) and use them for in vitro phosphorylation assays. While such approach has already been used, the proposed protocol is presented as being more reliable and straightforward than previously published methods. Altogether, this aims to provide a means for studying how Cdk1-dependent phosphorylation of mitotic substrates affect their structure and functions. In addition, they use the fission yeast homolog of Cks1, namely Suc1, which accelerate the phosphorylation rate of the substrates tested in this study.

The results are well-presented and the data are of high quality. However, some points need to be clarified or at least discussed to demonstrate that this protocol is promising and timely.

Major comments:

- The key goal of this method is to study how Cdk1-dependent phosphorylation of mitotic substrates affects their structure and functions. As the proposed protocol addresses this in vitro, it is critical to determine how these in vitro assays recapitulate in vivo situations. Not all Cdk1 target sites are phosphorylated in vivo, and this may depend on the concentration ratio of enzyme / substrate. As arbitrary concentration ratios are used for these in vitro assays, it remains possible that some mitotic Cdk1 substrates are hyperphosphorylated on TP or SP sites in these experiments, although these sites are not phosphorylated in vivo. For instance, from their previous studies, the authors show that their assay results in phosphorylation of at least two of the motifs present in mammalian condensin I. Does this imply that the in vitro assay does not fully recapitulate the in vivo situation? The authors could use substrates that are known to be phosphorylated at the time of mitosis but not on every potential target site (using for instance data from proteomic studies such as in fission yeast, Swaffer et al., Cell 2016) and compare this with their in vitro results. Comparing more thoroughly the results from the in vitro assay with the in vivo situation is key to validate the approach, as studying the structure and functions of substrates in a phosphorylated state that does not occur in vivo may not be relevant.

- As a control, the authors should test the phosphorylation state of substrates that only become phosphorylated by Cdk1 in association with different cyclins (for instance in G1 or S) and that are not phosphorylated by mitotic Cdk activity. If such substrates are also phosphorylated by M-CDK, the authors could then discuss cyclin specificity and, importantly, validate their approach not only for mitotic substrates but also for substrates that are key for other phases of the cell cycle.

- Different concentrations of M-CDK are used between the different figures. While the authors explain the use of lower concentrations in Fig. 2D, E compared to Fig. 2B, C, the use of yet a different concentration of M-CDK in Fig. 3 is not explained. Does it mean that different results are obtained by varying the concentrations of M-CDK? Do these changes have an impact on the potential in vitro hyper or hypo-phosphorylation of the substrates discussed above? If this is the case, the authors should discuss this, as it may be critical in vivo and should therefore be considered for interpreting in vitro data.

- Fig. 2D: The authors should be more cautious with their conclusions that the phosphorylated forms of H1.1 that they identify correspond to the five SP motifs present in the protein, unless they do a mutant analysis as performed for XD2-C (Fig. 3). In fact, as for XDC-2 (see comment below), the authors identify a phosphorylated form of H1.1 (arrow 3 in Fig. 2D) that is barely visible and does not change in intensity over time (in contrast to all other phosphorylated bands). More experiments would be necessary to convincingly demonstrate that this indeed corresponds to a phosphorylated H1.1 and that they identify all different phosphorylated forms on all 5 known target sites without ambiguity.

- In Fig. 3, the authors indicate that the substrate is phosphorylated on T1314, T1348 and T1353. However, while their data only show without ambiguity that T1314 and T1353 are phosphorylated, they conclude from the mutant analysis that all three TP motifs are targets of M-CDK. These results are not convincing. Indeed, on Fig. 3B-D, the authors indicate by arrows the presence of 3 phosphorylated bands on the Pho-tag gels. One of their arrows indicate a band that is barely if at all visible. In fact, while one would expect that the longer the substrate is incubated with M-CDK the more phosphorylated it becomes, that faint band (which in fact may be T1348, as this is the only residue that is not validated by an antibody directed against its phosphorylated form) does not increase in intensity over the time course of the experiment. The authors should either find a way to convincingly show that all three residues are indeed phosphorylated in their assay, or be more cautious with their conclusions.

Minor comments

- Why using Suc1 instead of the Xenopus or human protein?

- Materials and Methods: the authors should specify whether the exact same fragments (coordinates) are used for the human M-CDK than for the Xenopus M-CDK.

- Figure 1C: the blot anti-PSTAIR, as mentioned, identifies 2 bands, with the faster migrating band being the T161 phosphorylated form of Cdk1 (Fig. 1E). Comparing the signal with that of the L and F lines, it seems that the purified fractions are enriched in the T161 phosphorylated band. Could the authors comment on this?

- The phosphorylation rates presented in Fig. 2E are not further discussed. What is the relevance of this?

- Figure 2E: The error bars are not mentioned in the legend.

- Figure 3: What is the meaning of the colors of the different arrows in panels B and C? This should be explained in the legend. The “*” sign should also be explained in the legends of panels B and D.

- While the quantification in Fig. 3E validate the conclusion that the effect of Suc1 is dose-dependent, it is really not clear from the blot itself in Fig. 3D. Could the authors comment on this? Is the quality and robustness of the quantification of the phos-tag signal (and its normalization) sufficient to conclude on a dose-dependent effect that is not visible on the gel? Provided the goal of the method, does the idea that this is dose-dependent really relevant?

- Figure 3E: the error bars correspond to standard deviations. Here the authors do not evaluate the dispersion of phosphorylation status within a sample, but the reproducibility of their replicate experiments. They should therefore not use standard deviations but standard errors.

- I would suggest the authors to present the advantages of their methodology compared to previously published protocols in the introduction rather than in the conclusions.

- In the conclusions section, the authors discuss a “simplified protocol”. I would like to stress that the method that the author present, while potentially better than previous approaches, is not that simple either.

6. PLOS authors have the option to publish the peer review history of their article (what does this mean?). If published, this will include your full peer review and any attached files.

Reviewer #1: **Yes: **Daniel Fisher

Reviewer #2: No

---

## [Author Response · Author response to Decision Letter 0]

23 Jan 2024

Authors’ reply to reviewers

Academic Editor: Dr. Claude Prigent

(Comment 1, the former half) 

Better explain why your in vitro assay is better than existing assays; also find a way to demonstrate that the data obtained recapitulates certain in vivo situations by choosing the right substrates (see reviewer's comments). 

(Reply)

First of all, let us explain our original motivation for developing recombinant M-CDKs in the current study. We previously demonstrated that mitotic chromosomes can be reconstituted with only six purified proteins including a mitotically phosphorylated form of the native condensin I complex purified from Xenopus egg extracts (Shintomi et al, 2015, Nat Cell Biol). We also provided compelling evidence that M-CDK-dependent phosphorylation of condensin I is the sole post-translational modification required for chromosome assembly in this assay. However, the use of a native condensin I complex and a native M-CDK (purified from starfish oocytes) in this setup had an obvious technical limitation. If we can replace them with recombinant condensin I and recombinant M-CDK, it will become possible to address the important question of which residues (and which subunits) are the phosphorylation targets essential for mitosis-specific activation of condensin I. To this end, we have sought in vitro assay conditions that could recapitulate the multi-subunit and multi-site phosphorylation of condensin I observed in mitotic extracts of Xenopus eggs (nearly equivalent to an in vivo condition). Although some earlier analyses of condensin I used native or commercially available recombinant M-CDKs (Kimura et al 1998, Science; Takemoto et al, 2006, EMBO J; St-Pierre et al, 2009, Mol Cell), limitations in protein yield and reaction size prevent further extension to systematic analyses (e.g., using a panel of condensin mutants). Therefore, we decided to produce recombinant M-CDKs ourselves. 

In the current study, we have succeeded not only in producing this reagent with good yield and homogeneity but also in recapitulating the massive phosphorylation of condensin I in vitro. In particular, the generation of the phosphoepitopes at the C-terminus of CAP-D2 (namely, pT1314, pT1348, and pT1353; which are associated with the DNA supercoil inducing activity of condensin I [Kimura et al,1998, Science]) and the mobility shift of CAP-H (Hirano et al, 1997, Cell) provide compelling evidence for the recapitulation of physiologically relevant phosphorylation. Furthermore, while revising the current manuscript, we published another paper reporting that physiologically relevant multi-site phosphorylation of the CAP-D3 subunit of condensin II can be recapitulated in vitro using the same M-CDK preparation (Yoshida et al, 2024, Mol Biol Cell [PMID: 38088875]). Taken together, it would be fair to say that our assay is superior to existing assays for studying condensin phosphorylation. To clarify this point, we have newly added the following sentence in the text (page 21, lines 472-476).

In the final set of assays, multi-subunit and multi-site phosphorylation of the pentameric condensin I complex can be recapitulated by combining recombinant M-CDK with its cofactor Suc1. To our knowledge, this is the first report showing that physiologically relevant phosphorylation of condensin I can be achieved using recombinant M-CDK (Fig 4).

(Comment 1, the latter half) 

Establishing the classical kinetic parameters of a kinase would also be useful for comparison with other methods (see reviewer comment).

(Reply)

We fully appreciate this constructive comment. We have carried out an additional set of kinase assays to determine the Km and kcat (= Vmax/[E]) values of the recombinant X. tropicalis M-CDK for the histone H1.1 substrate (Fig 2F). The new result indicates that our M-CDK preparation catalyzes the phosphorylation of SP sites in H1.1 with an efficiency similar to those published previously (Table 2). We believe that this result will be helpful to potential readers in evaluating the quality of the M-CDK preparation. The corresponding sentences (page 14, lines 305-315) and citations ([8] Brown et al, 2015, Nat Commun and [24] Koivomagi et al, 2011, Mol Cell) have been added to the revised manuscript.

(Comment 2) 

Please explain why not use proteins from a single organism instead of using proteins from X. Tropicalis and human proteins for M-CDK with yeast Suc1.

(Reply)

As mentioned above, one of the long-term goals of our study is to recapitulate the mitotic chromosome assembly process using only recombinant proteins. More specifically, we are now trying to replace the phosphorylated native condensin I used in the original reconstitution assay with a combination of recombinant condensin I and M-CDK (established in this study). For this purpose, we have decided to produce M-CDKs and the condensin I complexes using frog (Xenopus) and human recombinant subunits. This is because the other recombinant proteins required for chromosome reconstitution (namely, Npm2, Nap1, FACT, core histones, and topo II) are derived from frogs or humans (Shintomi et al, 2015, Nat Cell Biol; Shintomi and Hirano, 2021, Nat Commun). In addition, proteins from species living at higher temperatures are generally known to have more stable structures. We have therefore produced M-CDK derived from Xenopus tropicalis (which inhabits the tropical zone with a water temperature of about 28ºC) rather than X. leavis (which inhabits the temperate or sun-tropical zone with a water temperature of about 21ºC).

That said, we should explain why we have used fission yeast Suc1 protein rather than frog or human Cks. One of the reasons is purely technical. We have attempted to produce recombinant human Cks2 using E. coli as a host. Although the final product showed good homogeneity, the yield was much lower than that of Suc1. Another reason is that the native cyclin B-Cdk1 complex from human cells is known to interact with Suc1 very tightly (Kusubata et al., 1993, J Biol Chem). Consistently, we have found that frog M-CDK composed of cyclin B1(�N,CS) and Cdk1(AF) can form a robust trimer together with fission yeast Suc1 in a structural model generated by AlphaFold2. Taking all into consideration, we think it is practical and reasonable to use a combination of fission yeast Suc1 and frog M-CDK for our purpose.

(Comment 3)

Please explain the use of different concentrations of M-CDK.

(Reply)

Let us explain how the concentrations of M-CDK were determined for phosphorylating each of the substrates. In the first set of experiments, we first titrated M-CDK concentrations against the linker histone H1.1 (Fig 2B, 2C, 2D, and 2G in the revised panel). As a result, we found that the lowest concentration (i.e., 10 nM) of M-CDK was sufficient for the subsequent quantitative analyses (Fig 2E and 2F). In the second set of experiments using the XD2-C substrate (Fig 3B, 3C, 3D, and 3E), the concentration of M-CDK was initially fixed at 20 nM and was maintained thereafter. Here we intended to slightly increase the enzyme concentration because XD2-C (a truncated form of XCAP-D2, which is one of the pentameric condensin I complex), unlike H1.1 (a full-length protein), might show somewhat poorer behaviors in solution.

The concentration of M-CDK used to phosphorylate condensin I was higher than those used to phosphorylate H1.1 and XD2-C. This was also determined by a careful titration experiment. When M-CDK was added at a concentration of 10 nM in a preliminary experiment, phosphorylation of CAP-C, - D2, and -H subunits was observed, but not to the full extent even in the presence of Suc1 (the corresponding result is attached here). We found that a relatively high concentration (50 nM) of M-CDK is required to phosphorylate condensin I to the level observed in mitotic cells. Therefore, we have presented the results using this concentration of M-CDK in the current manuscript and revised the corresponding part as follows (page 9, lines 201-202).

The concentrations of protein components in each assay, which were determined by preliminary titration experiments and technical considerations, are listed in Table 1.

(Comment 4)

Regarding multisite phosphorylation, please find a way to unambiguously control the identity of all phosphorylated sites (mutants or mass spectrometry) in H1.1 and XCAP-D2.

(Reply)

As part of our response to the reviewer’s comment, we have performed additional experiments as described below. We prepared an H1.1 mutant, in which all five serine residues in the SP motifs (S171, S184, S197, S213, and S232) were replaced with alanines, and then demonstrated that the resulting 5A mutant was barely phosphorylated in contrast to its wild type (Supporting information, S1 Fig). We did not try to identify phosphorylated sites by mass spectrometry because it was expected that the permutation of the primary structure surrounding the SP sites would make such an analysis difficult. Regarding the XD2-C substrate, we already addressed this issue in the original manuscript by using the 3A mutant (carrying the T1314A/T1348A/T1353 mutations) and Phos-tag SDS-PAGE (Fig 3B). Furthermore, in the revised manuscript, we have demonstrated that M-CDK hardly phosphorylates non-TP sites even in the presence of Suc1. It should be emphasized that this conclusion was drawn by three different methods: 1) Pro-Q diamond staining; 2) immunoblotting using anti-pT1314, anti-pT1348, and anti-pT1353 antibodies (following conventional SDS-PAGE); and 3) Phos-tag SDS-PAGE (Supporting information, S2 Fig). We are confident that these data are sufficient to list almost all phosphorylated sites in H1.1 and XD2-C. 

===

Reviewer #1: Dr. Daniel Fisher

(Comment, the first part)

The authors present an essentially methodological paper for the simple expression and purification of recombinant Xenopus and human cdk1-cyclin B complexes. They demonstrate that the kinases are active and can phosphorylate a recombinant condensin complex, and that addition of a small subunit (a CKS1 homologue) increases the phosphorylation rate. There are no particular surprises in these findings but the methodology and results are of high quality and the tools generated may be useful for the community.

(Reply) 

We are grateful to this reviewer for recognizing the potential impact of our manuscript on the research community.

(Comment, the second part)

It would be good if the authors provided a bit of basic extra characterisation to support their claim that the method is superior to previous methods - namely, by establishing kinetic parameters of the purified kinases (Km and Vmax for ATP and a substrate, e.g. histone H1), which would also allow to see if there are differences between the Xenopus and human complexes. 

(Reply)

We fully appreciate this constructive comment. To determine the kinetic parameters (i.e., Km, Vmax, and kcat) of the recombinant X. tropicalis M-CDK for the linker histone H1.1 substrate, we have carried out an additional kinase assay and demonstrated that our M-CDK preparation catalyzes the phosphorylation of SP sites in H1.1 with an efficiency similar to those published previously (Fig 2F newly added in the revised manuscript). For details, please see our reply to the Academic Editor’s comment 1, the latter half (page 2 in this authors’ reply). Furthermore, we already explained in the original manuscript that the final preparations of frog and human M-CDKs are unprecedentedly homogenous. Taken together, it is not an exaggeration to claim that our protocol developed here is superior to previous ones.

(Comment, the last part)

I also feel that to support the claim of efficient multisite phosphorylation of condensin complex, mass spectrometry should have been performed to identify phosphorylation sites. This is likely to also reveal that the recombinant kinases phosphorylate additional sites other than SP and TP sites, and might show whether Suc1 increases the rate of phosphorylation or allows additional sites to be phosphorylated.

(Reply)

Although a comprehensive identification of phosphorylation sites in the five-subunit condensin I complex would be highly informative, we believe that such efforts would be one of the future directions. We also consider that the functional impact of phosphorylation of given sites needs to be analyzed by combined functional assays. The current manuscript provides an important foundation for such future studies and is therefore worth publishing as is. 

Apart from the above point, we recognize that whether M-CDK phosphorylates non-SP/TP sites in condensin I subunits in the presence of Suc1 is a very important issue, as has been shown in various substrates by this reviewer and others (Al-Rawi et al, 2023 Cell Rep; Valverde et al 2023 Nat Commun). In the revised manuscript, we have provided additional data to demonstrate that little or no phosphorylation of non-SP/TP sites was detectable in the H1.1 or XD2-C at least under our assay conditions (Supporting information, S1 and S2 Figs). Of course, these data do not exclude the potential importance of non-SP/TP phosphorylation in different contexts. In the revised manuscript, we have newly added discussion about this point and citations as follows (Al-Rawi et al, 2023; Valverde et al 2023) (page 21, lines 469-472).

We provide a compelling set of data that the recombinant M-CDKs specifically phosphorylate SP and TP motifs in H1.1 and XD2-C substrates (Fig 2 and 3, S1 Fig and S2 Fig). However, our data do not exclude the possibility that Cdk1 also phosphorylates non-SP/TP sites in various cellular contexts [41, 42].

===

Reviewer #2

(Major comment 1) 

The key goal of this method is to study how Cdk1-dependent phosphorylation of mitotic substrates affects their structure and functions. As the proposed protocol addresses this in vitro, it is critical to determine how these in vitro assays recapitulate in vivo situations. Not all Cdk1 target sites are phosphorylated in vivo, and this may depend on the concentration ratio of enzyme / substrate. As arbitrary concentration ratios are used for these in vitro assays, it remains possible that some mitotic Cdk1 substrates are hyperphosphorylated on TP or SP sites in these experiments, although these sites are not phosphorylated in vivo. For instance, from their previous studies, the authors show that their assay results in phosphorylation of at least two of the motifs present in mammalian condensin I. Does this imply that the in vitro assay does not fully recapitulate the in vivo situation? The authors could use substrates that are known to be phosphorylated at the time of mitosis but not on every potential target site (using for instance data from proteomic studies such as in fission yeast, Swaffer et al., Cell 2016) and compare this with their in vitro results. Comparing more thoroughly the results from the in vitro assay with the in vivo situation is key to validate the approach, as studying the structure and functions of substrates in a phosphorylated state that does not occur in vivo may not be relevant.

(Reply)

We should sort out the issue of to what extent our in-vitro phosphorylation assays recapitulate in-vivo situations. It is not appropriate to discuss this issue on H1.1 and XD2-C because each substrate is present as part of a supramolecular structure (i.e., a chromatosome [a nucleosome that binds to a linker histone] and the condensin I complex) under physiological conditions, unlike the phosphorylation assay conditions used in this study. It is therefore productive to focus on condensin I phosphorylation.

As this reviewer pointed out, not every potential target site in condensin I may always be phosphorylated. It is also possible that non-physiological target sites are modified by M-CDK in our in-vitro assays. We fully agree with this reviewer’s concerns, but rigorously addressing these issues is clearly beyond the scope of the current study. Furthermore, although the literature this reviewer mentioned (Swaffer et al, 2016, Cell) provided data regarding a variety of CDK substrates in fission yeast, the primary structures of these proteins are little conserved in vertebrates. For this reason, even if our data regarding vertebrate condensins were compared with these fission yeast data, it would not be straightforward to make a fruitful interpretation. On the other hand, we have mainly focused on the multi-site phosphorylation of condensin I subunits, which is originally characterized in mitotic extracts of Xenopus eggs (nearly equivalent to an in vivo condition), and we are confident that the data presented in the current paper are sufficient to demonstrate that these physiologically relevant phosphorylation events can be reproduced using recombinant M-CDK. For details, please see our argument (as a reply to the Academic Editor’s comment 1).

(Major comment 2) 

As a control, the authors should test the phosphorylation state of substrates that only become phosphorylated by Cdk1 in association with different cyclins (for instance in G1 or S) and that are not phosphorylated by mitotic Cdk activity. If such substrates are also phosphorylated by M-CDK, the authors could then discuss cyclin specificity and, importantly, validate their approach not only for mitotic substrates but also for substrates that are key for other phases of the cell cycle.

(Reply)

We respectfully disagree with this comment. Contrary to the reviewer’s suggestion, it is highly unlikely that frog or human Cdk1 (used in the current study) binds to either G1 cyclins or S cyclins. The reviewer may be somewhat confused because budding yeast Cdk1 does indeed bind to both G1/S cyclins and M cyclins.

During mitosis, many “authentic” substrates of M-CDK are known to undergo massive phosphorylation, at multiple sites in individual proteins and in multiple subunits within protein complexes. These trends are less pronounced for substrates that are phosphorylated outside of mitosis or for substrates that are phosphorylated in mitosis by kinases other than M-CDK. The purpose of the current study is to develop a protocol to recapitulate M-CDK-mediate multi-site and multi-subunit phosphorylation of authentic substrates (e.g., the condensin I complex) in vitro. We are confident that this study will pave the way for investigating how the functions and structures of these substrates are controlled by phosphorylation under such near-physiological assay conditions. Although it is formally possible to perform a phosphorylation assay using a combination of a “non-authentic” substrate and M-CDK, we do not believe that we could obtain any biologically meaningful information from such an experiment. 

(Major comment 3) 

Different concentrations of M-CDK are used between the different figures. While the authors explain the use of lower concentrations in Fig. 2D, E compared to Fig. 2B, C, the use of yet a different concentration of M-CDK in Fig. 3 is not explained. Does it mean that different results are obtained by varying the concentrations of M-CDK? Do these changes have an impact on the potential in vitro hyper or hypo-phosphorylation of the substrates discussed above? If this is the case, the authors should discuss this, as it may be critical in vivo and should therefore be considered for interpreting in vitro data.

(Reply)

We have already explained step by step how we determined the concentration of M-CDK for each of the substrates, as a reply to Academic Editor’s comment #3.

Keeping all of our results and the abovementioned argument in mind, I would like to answer the following specific questions:

- Does it mean that different results are obtained by varying the concentrations of M-CDK? 

Yes. For H1.1, the catalytic rate changes dependently on M-CDK concentrations (Fig 2G). On the other hand, it is most likely different concentrations of M-CDK and Suc1 yield qualitative and quantitative differences in phosphorylation of condensin I’s subunits.

- Do these changes have an impact on the potential in vitro hyper or hypo-phosphorylation of the substrates discussed above?

The answer is yes (at least for condensin I). So far, we have learned that changing the concentration of M-CDK phosphorylates condensin I at different levels in a highly reproducible manner. In addition, our preliminary results from the chromosome reconstitution assay indicate that the functions of condensin I depend on its phosphorylation levels. The relationship between phosphorylation levels and biochemical activities will be an important subject for future studies. 

(Major comment 4) 

Fig. 2D: The authors should be more cautious with their conclusions that the phosphorylated forms of H1.1 that they identify correspond to the five SP motifs present in the protein, unless they do a mutant analysis as performed for XD2-C (Fig. 3). In fact, as for XDC-2 (see comment below), the authors identify a phosphorylated form of H1.1 (arrow 3 in Fig. 2D) that is barely visible and does not change in intensity over time (in contrast to all other phosphorylated bands). More experiments would be necessary to convincingly demonstrate that this indeed corresponds to a phosphorylated H1.1 and that they identify all different phosphorylated forms on all 5 known target sites without ambiguity.

(Reply)

We must admit that our explanation of this point in the original text was incomplete. We followed this reviewer’s helpful advice and performed the following additional set of experiments. When all five serine residues in the SP motifs (S171, S184, S197, S213, and S232) of H1.1 were replaced with alanines, the resulting 5A mutant was barely phosphorylated by M-CDK (Supporting information, S2 Fig). It is therefore reasonable to conclude that M-CDK specifically phosphorylates some or all of the five SP motifs, but not other residues, in our assay. 

As pointed out by this reviewer, we must admit that one out of the five phosphorylated forms of H1.1 (indicated by arrow 3 in Fig 2D) was rather weak. We did not use mass spectrometry to identify all the phosphorylation sites because the permutations of the primary structure around the SP sites were expected to make such attempts technically difficult. Therefore, we cannot completely exclude the possibility that any one of the five sites was not phosphorylated. In the revised manuscript, to clarify this point, the following sentence has been added (page 15, lines 340-341). 

However, we cannot exclude the possibility that any one of the five sites was not phosphorylated in our assay. 

(Major comment 5) 

In Fig. 3, the authors indicate that the substrate is phosphorylated on TP siteT1314, T1348 and T1353. However, while their data only show without ambiguity that T1314 and T1353 are phosphorylated, they conclude from the mutant analysis that all three TP motifs are targets of M-CDK. These results are not convincing. Indeed, on Fig. 3B-D, the authors indicate by arrows the presence of 3 phosphorylated bands on the Pho-tag gels. One of their arrows indicate a band that is barely if at all visible. In fact, while one would expect that the longer the substrate is incubated with M-CDK the more phosphorylated it becomes, that faint band (which in fact may be T1348, as this is the only residue that is not validated by an antibody directed against its phosphorylated form) does not increase in intensity over the time course of the experiment. The authors should either find a way to convincingly show that all three residues are indeed phosphorylated in their assay, or be more cautious with their conclusions.

This reviewer’s comment is correct. We admit the data presented in the original manuscript were insufficient to conclude that M-CDK phosphorylates all three TP sites (T1314, T1348, and T1353) of XD2-C. We consider that the following two factors made our conclusion less than fully convincing: (1) the lack of immunoblot analysis using anti-pT1348; (2) the difficulty in precisely counting the number of phosphorylated residues using Phos-tag gel. With these factors in mind, we performed a new experiment in which either the wild-type (WT) or the 3A mutant of XD2-C was incubated with M-CDK (Supporting information, S2 Fig, newly added). First, in addition to pT1314 and pT1353, pT1348 was found to be positive on XD2-C WT but not on 3A. Second, using not only Phos-tag SDS-PAGE but also conventional SDS-PAGE followed by Pro-Q diamond staining, we found little or no evidence for phosphorylation on the 3A mutant. Furthermore, we confirmed that the same was true under a condition where Suc1 stimulates multi-phosphorylation reactions. Collectively, we are almost certain that all of the three TP sites, but not other sites, of the XD2-C substrate are specifically phosphorylated by M-CDK under our experimental setup. 

(Minor comment 1)

Why using Suc1 instead of the Xenopus or human protein?

(Reply)

We have explained the reason in our reply to the Academic Editor’s comment 2. For details, please see page 3 in this document. Taking this into consideration, we think that it is practical and reasonable to use a combination of fission yeast Suc1 and frog M-CDK in the current study.

(Minor comment 2)

Materials and Methods: the authors should specify whether the exact same fragments (coordinates) are used for the human M-CDK than for the Xenopus M-CDK.

(Reply)

We have provided detailed information about the cDNAs encoding engineered versions of human cyclin B1 and Cdk1 used in the current study and their cloning procedures in the Materials and Methods section (see pages 5-6, lines 113-126 in the revised text). 

(Minor comment 3)

Figure 1C: the blot anti-PSTAIR, as mentioned, identifies 2 bands, with the faster migrating band being the T161 phosphorylated form of Cdk1 (Fig. 1E). Comparing the signal with that of the L and F lines, it seems that the purified fractions are enriched in the T161 phosphorylated band. Could the authors comment on this?

(Reply)

The reviewer is right. Our interpretation of the results is as follows: Endogenous CAK preferentially phosphorylates Cdk1 complexed with cyclin B over free Cdk1 in insect cells. Although the cell lysates and eluates from the first column contain both cyclin B-Cdk1 and free Cdk1, the second column enriches the former fraction of Cdk1 (i.e., cyclin B-Cdk1). This is the reason why the purified sample is apparently enriched in the pT161-positive band. However, we are afraid that the inclusion of this argument may interfere with the overall logical flow and the readability of the current manuscript. For this reason, we would like to refrain from mentioning this point in the text.

(Minor comment 4)

The phosphorylation rates presented in Fig. 2E are not further discussed. What is the relevance of this?

(Reply)

Please let us explain our intention in presenting Fig 2E. In the original manuscript, we simply wanted to show that the phosphorylation rate can be measured from the Phos-tag SDS-PAGE data. During the revision process, an additional important meaning was added to this panel. We have carried out a classical kinetics assay to address one of the comments from another reviewer. This assay required the measurement of the phosphorylation rates (equivalent to the velocities) under a variety of different concentrations. In the revised version, we have left Fig 2E as a description of how to measure the reaction rates and then provided the new data set from the kinetics analysis as Fig 2F.

Note that because the reaction rate measurement has been also done in Fig 3E, Fig 2E serves as a validation of this methodology.

(Minor comment 5)

Figure 2E: The error bars are not mentioned in the legend.

(Reply)

As the reviewer pointed out in Fig 3E (minor comment 8), we should have used “standard errors” rather than standard deviations in Fig 2E. Therefore, in addition to specifying what the error bars indicate in the legend, we have also revised the actual graph panel used in this figure and the corresponding values in the text.

(Minor comment 6)

Figure 3: What is the meaning of the colors of the different arrows in panels B and C? This should be explained in the legend. The “*” sign should also be explained in the legends of panels B and D.

(Reply)

We appreciate this comment. Our original motivation for using the arrows in different colors was to indicate which band in one panel corresponds to which band in another panel. However, this attempt was not easy to understand without additional explanation and had little relevance to the evaluation of the results. We have therefore deleted the arrows and replaced them with brackets that collectively indicate the bands of phosphorylated XD2-C at different levels in the revised version of Fig 3B and 3C.

We apologize for the confusion caused by the “*” sign in the original manuscript. This indicated a degradation product of XD2-C or its abortively translated polypeptide (most likely equivalent to the MBP moiety alone as judged by its electrophoretic mobility). We have properly explained this in the revised manuscript.

(Minor comment 7)

While the quantification in Fig. 3E validate the conclusion that the effect of Suc1 is dose-dependent, it is really not clear from the blot itself in Fig. 3D. Could the authors comment on this? Is the quality and robustness of the quantification of the phos-tag signal (and its normalization) sufficient to conclude on a dose-dependent effect that is not visible on the gel? Provided the goal of the method, does the idea that this is dose-dependent really relevant?

Although our quantification clearly shows a significant difference between 0.1 uM and 1.0 uM Suc1, the titration in this experiment was admittedly very rough. Our primary goal here was to show that the addition of Suc1 increases the efficiency of the phosphorylation reaction, and we have no strong intention of claiming the dose dependency of Suc1. In the revised manuscript, we have removed “in a dose-dependent manner” from the corresponding sentence. 

(Minor comment 8)

Figure 3E: the error bars correspond to standard deviations. Here the authors do not evaluate the dispersion of phosphorylation status within a sample, but the reproducibility of their replicate experiments. They should therefore not use standard deviations but standard errors.

(Reply)

This comment is reasonable. As the reviewer pointed out, it would be better to show “standard errors” rather than standard deviations. We have thus corrected the error bars in the graph panels and the figure legend accordingly. In addition to Fig 3E, we realized that Fig 2E should also be revised, and we have done so.

(Minor comment 9)

I would suggest the authors to present the advantages of their methodology compared to previously published protocols in the introduction rather than in the conclusions.

(Reply)

Although we do not fully agree with this suggestion, we do agree that it is helpful to briefly state the advantages of our method in the introduction. We have therefore rephrased the sentence as follows (page 3, lines 62-66). 

In this paper, we report a modified protocol for the production of recombinant cyclin B-Cdk1 at a high homogeneity. The resultant kinase can specifically phosphorylate cyclin-dependent kinase consensus motifs (SP and TP sites) of substrates [23] at a catalytic efficiency better than budding yeast recombinant M-CDK previously reported [24].

We would like to keep the corresponding discussions provided in the original manuscript as they are.

(Minor comment 10)

In the conclusions section, the authors discuss a “simplified protocol”. I would like to stress that the method that the author present, while potentially better than previous approaches, is not that simple either.

(Reply)

This reviewer’s comment is a reasonable one. We have thus deleted the word “simplified” from the corresponding sentence.

---

## [Decision Letter · Decision Letter 1]

5 Feb 2024

Recombinant cyclin B-Cdk1-Suc1 capable of multi-site mitotic phosphorylation in vitro

PONE-D-23-30585R1

Dear Dr. Shintomi,

We’re pleased to inform you that your manuscript has been judged scientifically suitable for publication and will be formally accepted for publication once it meets all outstanding technical requirements.

Kind regards,

Claude Prigent

Academic Editor

PLOS ONE

Additional Editor Comments (optional):

Reviewers' comments:

Reviewer's Responses to Questions

**Comments to the Author**

1. If the authors have adequately addressed your comments raised in a previous round of review and you feel that this manuscript is now acceptable for publication, you may indicate that here to bypass the “Comments to the Author” section, enter your conflict of interest statement in the “Confidential to Editor” section, and submit your "Accept" recommendation.

Reviewer #1: All comments have been addressed

2. Is the manuscript technically sound, and do the data support the conclusions?

Reviewer #1: Yes

3. Has the statistical analysis been performed appropriately and rigorously? 

Reviewer #1: Yes

4. Have the authors made all data underlying the findings in their manuscript fully available?

Reviewer #1: Yes

5. Is the manuscript presented in an intelligible fashion and written in standard English?

Reviewer #1: Yes

6. Review Comments to the Author

Reviewer #1: (No Response)

7. PLOS authors have the option to publish the peer review history of their article (what does this mean?). If published, this will include your full peer review and any attached files.

Reviewer #1: **Yes: **Daniel Fisher

---

## [Editor Report · Acceptance letter]

12 Mar 2024

PONE-D-23-30585R1 

PLOS ONE

Dear Dr. Shintomi, 

I'm pleased to inform you that your manuscript has been deemed suitable for publication in PLOS ONE. Congratulations! Your manuscript is now being handed over to our production team.

Kind regards, 

on behalf of

Dr. Claude Prigent 

Academic Editor

PLOS ONE